# *Cherax quadricarinatus* Resistant to *Chequa iflavirus*: A Pilot Study

**DOI:** 10.3390/microorganisms11030578

**Published:** 2023-02-24

**Authors:** Geetika Nambiar, Leigh Owens, Jennifer Elliman

**Affiliations:** College of Public Health, Medical and Veterinary Sciences, James Cook University, 1 Solander Road, Townsville, QLD 4811, Australia

**Keywords:** *Cherax quadricarinatus*, redclaw crayfish, *Chequa iflavirus* resistance, Athtab bunyavirus

## Abstract

High mortalities of redclaw crayfish (*Cherax quadricarinatus*) were reported from northern Queensland farms, mainly attributed to two viruses, *Chequa iflavirus* and Athtab bunyavirus. From a research population of redclaw crayfish with these pre-existing viral infections, five individuals were found uninfected by *Chequa iflavirus* but infected with Athtab bunyavirus. A pilot study was designed to examine if progeny crayfish from this cohort were resistant to infections by *Chequa iflavirus*. Two experiments measured changes in viral load with RT-qPCR. Seven donors, four negative controls and six crayfish injected with a purified virus or saline were used. In Experiment 1, the purified viral inoculum was injected into the crayfish, and they were bled 14 days post-injection (dpi). In Experiment 2, haemolymph containing the viruses was injected into the same crayfish and they were bled at 24 hpi, 48 hpi, 7 dpi and 14 dpi. In Exp. 1, the crayfish cleared *Chequa iflavirus* infections within 14 dpi, while in Exp. 2, it was within 24 hpi. One mortality was observed, but that crayfish had cleared the virus before dying. The number of copies of Athtab bunyavirus and the weights of the crayfish did not differ significantly (*p* > 0.05) between the control and injected crayfish. Histology of crayfish all showed that the haemolymph vessels were clear of granulomas, suggesting no bacterial involvement. There was no melanisation in the gill tissue of control crayfish, but it was prominent in virus-injected crayfish. Neither group had haemocytic infiltration of the muscle fibres. Anti-viral immune mechanisms of RNA interference and *Cherax quadricarinatus* Down Syndrome Cell Adhesion Molecule (DSCAM) are hypothesised to be involved in viral clearance. We conclude that these crayfish were resistant to *Chequa iflavirus* infections and could be commercially exploited by aquaculturists as a nuclear breeding stock if numbers are increased over time.

## 1. Introduction

*Cherax quadricarinatus* (von Martens, 1868), redclaw crayfish, is a freshwater decapod crustacean native to northern Australia and southern Papua New Guinea [1,2,3]. Redclaw crayfish were first cultured in Australia in the mid-1980s [4]. However, it has been translocated to other countries due to its popularity in the aquaria trade and aquaculture [5]. Currently, Australia, Ecuador, Uruguay and Mexico are major producers of redclaw crayfish [6].

FAO [6] reports that between 2010 and 2016, an estimated 1323 tonnes of redclaw crayfish was produced, but production per annum is highly volatile, including Queensland, Australia’s largest producer, where it decreased 25% in value for the 2017–2018 financial year, down from 2016 to 2017 [7]. In Australia, the dismal statistics can be partially attributed to a lack of government incentives that have affected investment in redclaw crayfish aquaculture [6]. Additionally, diseases have also affected redclaw crayfish production. For example, from 1999 to 2000, a northern Queensland (NQ) farm reported up to 96% mortalities due to outbreaks of a parvo-like virus [8], later identified as *Cherax quadricarinatus* Aquambidensovirus [9]. Similarly, Sakuna et al. [10] reported that in 2015, up to 65% mortality of redclaw crayfish from one NQ farm due to the viruses described herein.

*Chequa iflavirus* is the first iflavirus recorded in crustacea [10]. As a positive sense, ssRNA virus with icosahedral symmetry belongs phylogenetically to the family *Iflaviridae*, order *Picornavirales*. The virus was associated with stress-triggered mortalities in northern Queensland farms and is potentially the cause of brittle muscle fibres. Sakuna et al. [11] conducted therapeutic trials and were successful in reducing the viral copy numbers in redclaw. The research group also attempted to prove Rivers’ postulates for the virus using house cricket (*Acheta domesticus*) as their model, an uninfected experimental animal, but they were unsuccessful [12]. *A. domesticus* was trialled because there was a complete lack of *Chequa iflavirus* and Athtab bunyavirus-free redclaw crayfish because there is in ovo infection by these viruses [13]. This was also the same problem faced by this present study.

In the therapeutic trials conducted by Sakuna et al. [11], five redclaw crayfish did not show infections by *Chequa iflavirus* and were sequentially tested 5–6 times by RT-qPCR to confirm this. The crayfish remained uninfected despite them being housed in the same tanks as *Chequa iflavirus*-infected crayfish. Those authors recommended using this small sub-population (five out of 140 redclaw tested) for propagation and subsequently using them for selective breeding programs. From those five uninfected redclaw, a small colony of crayfish was developed over two years, which forms the basis of this pilot study. As the crayfish were previously uninfected even after co-habitation with infected crayfish and were likely the progeny of *Chequa iflavirus*-infected parents, the hypothesis of the present study was that this small sub-population is somehow resistant to being infected by *Chequa iflavirus*.

Hence, the aims of this study were as follows: (1) to confirm if the *Chequa iflavirus*-free population was resistant to *Chequa iflavirus* infections; (2) as an unavoidable corollary, to observe the effect of infecting the redclaw with pre-existing Athtab bunyavirus with more Athtab bunyavirus.

## 2. Methods and Materials

### 2.1. Crayfish

*Cherax quadricarinatus* (*n* = 26) were sourced from two isolated populations held at James Cook University (JCU). Twelve were viral donor crayfish from Population 1, and fourteen were Chequa-free crayfish from Population 2. Population 1 was naturally infected with *Chequa iflavirus* and Athtab bunyavirus without overt signs of diseases. This population has been maintained since 1995. Individual crayfish from Population 1 were examined intermittently since the original experiment by Sakuna et al. [10]. to check for *Chequa iflavirus* and Athtab bunyavirus load, which were found to be consistently high (10^5^–10^8^ copies/µL). Population 2 crayfish were infected with Athtab bunyavirus but free from *Chequa iflavirus* infection. These were bred from a small Chequa-free subset of Population 1.

For experiment 1, seven crayfish from Population 1 were relocated to a separate 1000 L plastic tank, but all crayfish died 3 weeks after relocation, exceeding the 65% mortality reported in Sakuna et al. [10], so muscle tissue was used to develop a viral inoculum. For experiment 2, five crayfish were directly bled at the Population 1 husbandry tanks for the viral inoculum.

The fourteen crayfish from Population 2 had a mean weight of 33.1 ± 12.4 g. Four were selected as to be free-roaming scavengers in the 1000 L tanks for both experiments, and the remaining 10 crayfish from Population 2 were individually housed in circular experimental pots made of plastic mesh and shade cloth, 17.5 cm in diameter and 22.5 cm in height, placed in 1000 L plastic tanks. All crayfish were fed chicken pellets, green beans and mackerel weekly on alternate days and were observed during feeding. There was a 12 h light: 12 h dark photoperiod and the temperature was maintained at 26 °C. All crayfish were allowed to acclimate for at least 2 weeks before experimentation. All crayfish were treated ethically according to the guidelines of the Royal Society for the Prevention of Cruelty to Animals. All crayfish that died before, during and after the experiment was cut longitudinally. One half was stored in Davidson’s fixative, and the other half was stored at −20 °C until required.

### 2.2. Viral Extract

One donor crayfish with the highest copy number of *Chequa iflavirus* (2.28 × 10^6^/µL) and the lowest copy number of Athtab bunyavirus (3.57 × 10^6^/µL) was chosen. Tail muscle, 5.6 g, was extracted, cut into five mm^3^ pieces and homogenised in 35 mL TNE buffer in a Stomacher^®^ 400 Circulator (Seward Ltd., Worthing, UK) at 8× *g* for 2 min. The homogenated tissue was centrifuged at 3300× *g* for 30 min at 4 °C to remove coarse debris using Sorvall RC6 Plus Superspeed Centrifuge (Thermo Fisher Scientific, Waltham, MA, USA). The supernatant was further clarified by centrifugation at 15,200× *g* for 30 min at 4 °C. The supernatant was extracted and ultracentrifuged at 130,000× *g* for 2 h at 4 °C using an SW 55 Ti rotor in an Optima L-90K Ultracentrifuge (Beckman Coulter, Brea, CA, USA). The pellet was resuspended in 800 PBS and the inoculum was aliquoted for injections and RNA extraction. All tubes were stored at −80 °C until required. RT-qPCR revealed that the viral inoculum had 1.6 × 10^6^ copies/µL of *Chequa iflavirus* and 3.51 × 10^6^ copies/µL of Athtab bunyavirus. Injections were administered intramuscularly between the first and second abdominal segments on the dorsal side.

### 2.3. Bleeding, Weighing and RNA Extraction

Sterile 1 mL BD Luer-Lok™ syringes with a 26 g needle were used to extract 100 µL haemolymph from the crayfish. All syringes were first filled with 10 µL sodium citrate as an anticoagulant. The crayfish were immersed in ice-cold water till they were unresponsive and then bled by inserting the syringe under the last pereiopod. In Exp. 1, the crayfish were patted dry using a paper towel, placed on a scale and weighed. RNA was extracted using 100 µL of haemolymph unless stated otherwise, using Total RNA Purification Kit (NorgenBiotek^®^, Thorold, ON, Canada), following the manufacturer’s instructions.

### 2.4. Reverse Transcription

cDNA was produced using Tetro cDNA Synthesis Kit (Bioline, London, UK) per the manufacturer’s instructions using PCR-specific primers (Table 1). The reaction was facilitated using a GeneTouch thermal cycler (Bioer Technology Co. Ltd., Hangzhou, China). Cycling conditions were as follows: 45 °C/30 min, 85 °C/5 min and held at 4 °C. Primers for *Chequa iflavirus* and Athtab bunyavirus were obtained from Sigma-Aldrich (St. Louis, MO, USA) according to those published by Sakuna et al. [10,11] (Table 1).

### 2.5. Quantitative PCR (q-PCR)

A 20 µL reaction mix for *Chequa iflavirus* contained 10 µL of 2× SensiFast SYBR No ROX Buffer, 1.6 µM of each primer, 2 µL of the DNA template and nuclease-free water. The cycling conditions were as follows: 95 °C/10 min followed by 40 cycles of 90 °C/5 s, 59 °C/10 s and 72 °C/10 s. For Athtab bunyavirus, a 20 µL reaction mix contained 10 µL of 2× SensiFast SYBR No ROX Buffer, 2 µM of each primer, 2 µL of the template (made using the protocol in 2.4) and nuclease-free water. The cycling conditions were as follows: 95 °C/3 min followed by 40 cycles of 95 °C/5 s, 59 °C/10 s and 72 °C/10 s. The reaction was facilitated using a Rotor-Gene Q thermal cycler (QIAGEN, Hilden, Germany), and the primers used were the same as those used for RT-PCR (Table 1). Standards made by Sakuna et al. during their 2017 and 2018 studies [10,11,12] were used as positive controls. Quantitative and melt curve analysis was performed at the end of each PCR run using Rotor-Gene Q Series Software 2.3.1 (QIAGEN). When necessary, PCR amplicons were visualised on 2% agarose gel with GelRed™ (Biotium, Fremont, CA, USA) and run using 1× TAE buffer. GeneRuler 100 bp Plus DNA ladder (Thermo Fisher Scientific) and HyperLadder 100 bp Plus (Bioline) were used, and the gel was run for 40 min at 200 v volts before visualisation using InGenius3 Gel Imaging System (Syngene, Frederick, MD, USA).

### 2.6. Experimental Design

#### 2.6.1. Pre-Experiment Screening

Before the commencement of the experiments, viral titres of negative control, donor and viral extract injected crayfish were established for *Chequa iflavirus* and Athtab bunyavirus, respectively. From the small *Chequa iflavirus*-free population, juvenile and adult crayfish of moderate size (~33 g) were selected (10 crayfish) and screened for *Chequa iflavirus* and Athtab bunyavirus to ensure that the crayfish had no infections by the former. These crayfish were then used in the experiments. Seven positive donor crayfish were bled after one week of acclimatisation in their tank to identify which crayfish would be used to purify *Chequa iflavirus*. After their subsequent death, tissue was extracted from all crayfish and tested again to check the viral titre for *Chequa iflavirus* and Athtab bunyavirus prior to selection for production of viral extract (Section 2.2).

#### 2.6.2. Experiment 1

Ten crayfish from Population 2 were assigned as experimental animals, while the remaining 4 were added to the tanks as free-range scavenging crayfish but were not included in the analysis during Experiment 1. Six of the ten crayfish were assigned to the viral manipulation tank and four to the negative control tank. The assignment was based on selecting two crayfish of similar weight and using one for viral infection and one as a weight-matched PBS-injected negative control without further viral exposure. The extra two unmatched crayfish in the viral manipulation tank were for a PBS injection control (PC) and an uninjected husbandry control (HC), intended to see if *Chequa iflavirus* was being transmitted via the habitation water. Fifty µL and 100 µL (to observe any dose-related protein shock response) of viral extract (Section 2.2) were injected into two crayfish each from the viral injection group (Table 2). For statistical analyses, the control group consisted of all the crayfish that were injected with PBS (4 crayfish in the negative control tank and the PC). The experiment began on day 0 when the crayfish were injected. Bleedings were planned every 14 dpi until day 42; however, Exp. 1 concluded after the first bleeding on day 14, after negative iflavirus titres. The haemolymph was stored at −80 °C until required, and 50 µL of haemolymph was used to extract RNA.

#### 2.6.3. Experiment 2

This experiment was designed after the results from Exp. 1. The aim of Exp. 2 was to identify how quickly the crayfish cleared *Chequa iflavirus*. Considering the rare possibility that the virus was inactivated while making the viral extract in Exp. 1, Experiment 2 utilised untreated haemolymph. The haemolymph was collected from five randomly selected crayfish belonging to the original donor Population 1 and pooled. Injections were administered the same day the haemolymph was collected to avoid freeze–thaw effects. An amount of 100 µL of viral-laden haemolymph was injected directly into the same crayfish as Exp. 1 (Table 3), with the exception that the husbandry control (HC) from Experiment 1 was used as a replacement experimental animal for I1 that died and the procedural control (PC); it was used as a husbandry control (HC) because it had already been injected with PBS in Experiment 1. RT-qPCR revealed that the haemolymph had 3.33 × 10^7^ copies/µL of *Chequa iflavirus* and 7.29 × 10^6^ copies/µL of Athtab bunyavirus. Exp. 2 began on day 0 when viral and control injections were administered. All crayfish were planned to be bled 24 hpi, 48 hpi, 7 dpi and 14 dpi after the injections, and the haemolymph was stored at −80 °C until required.

### 2.7. Histopathology

Histology was performed after Exp. 2 on one crayfish from the control tank and all crayfish from the experimental group, including HC and free-ranging scavenger crayfish (the scavengers remained free ranging in the tank during both experiments). In brief, gill, hepatopancreatic and tail muscle tissues were extracted from one-half of the crayfish stored in Davidson’s fixative after being transferred to 70% ethanol. Tissues were cropped into histocassettes and washed in ethanol to dehydrate and xylene for clearing. Subsequently, they were embedded in paraffin wax and stained using H&E. Sections were visualised under a light microscope, and photographs were taken using Olympus DP21 digital camera.

### 2.8. Statistical Analyses

Viral copy number/µL was calculated from the RT-qPCR and compiled in MS Excel ver. 16.36 (Microsoft, Redmond, WA, USA). As the sample size was small, the study acknowledges the low statistical power. Despite the viral copy number being log_10_ transformed, some groups were still not normally distributed; hence, non-parametric tests were used [14].

For Exp. 1, the Wilcoxon signed-rank test for unpaired data and the Wilcoxon rank-sum test for paired data were performed.

For Exp. 2, the Kruskal–Wallis rank-sum test was performed, followed by a pairwise comparison using the Wilcoxon rank-sum test as a post hoc test.

For the weight of the crayfish, Welch two sample *t*-test and Paired *t*-test were performed after passing normality testing using Shapiro–Wilk test. All statistical tests were performed on R ver. 3.6.3 (R Core Team, Vienna, Austria, 2020) and considered significant at *p* < 0.05.

## 3. Results

### 3.1. Experiment 1

#### 3.1.1. *Chequa iflavirus*

The virus-injected crayfish cleared *Chequa iflavirus* within 14 days of being infected. No copies of *Chequa iflavirus* were detected by RT-qPCR (Figure 1), including in PC and HC. The melt analysis indicated that the injected crayfish host amplicons had an average melting temperature of 86.7 ± 0.25 °C while the *Chequa iflavirus* standards had a concentration of 10^7^ and an average melting temperature of 80.23 ± 0.05 °C (Figure 1b). The temperature threshold was set to 75.4 °C, and the amplification efficiency for the assay was 0.90. Although one crayfish (crayfish I1) from the inoculum group died during the experiment, RT-qPCR using its tissue revealed that the crayfish had cleared *Chequa iflavirus* prior to dying. Negative-control crayfish were also not infected by the *Chequa iflavirus* when tested. The RT-qPCR products were visualised using gel electrophoresis to confirm the absence of *Chequa iflavirus*. After receiving the results from the first bleed post-infection, subsequently planned bleedings were terminated as the aim was fulfilled.

All planned donor crayfish died 3 weeks after relocation. The dead crayfish were stored at −20 °C, and subsequently, RNA was extracted from the tail muscle. RT-qPCR showed that the copy numbers of *Chequa iflavirus* and Athtab bunyavirus were not significantly different (*p* > 0.05) at the time of their death when compared to previous samples collected when the crayfish were introduced to a new tank.

#### 3.1.2. Athtab Bunyavirus

Although Athtab bunyavirus copies did not change significantly over the course of Exp. 1, RT-qPCR results indicated a modest reduction in bunyaviral copies (Figure 2).

There was no significant difference in the bunyaviral load in the injection and control group before injections (*p* = 0.1905) or after injections (*p* = 0.9048). Furthermore, no significant difference in copies was observed in the inoculum group before and after injecting the viral inoculum (*p* = 0.125). All analyses included the dead crayfish I1 from the viral injection tank, which had comparable copies of Athtab bunyavirus to other crayfish as confirmed by RT-qPCR. Pre-injection, viral copies in most crayfish were in the range of 10^8^ copies/µL and remained within that range except for I3, which saw a decrease in copies from 3.4 × 10^8^ to 2.1 × 10^7^ copies/µL (Figure 2) and PC where copies decreased from 7.2 × 10^8^ to 2.4 × 10^7^. While the former was injected with viral inoculum, the latter was injected with PBS, and as there are no significant changes, they are probably just evidence of the Regression To the Mean (RTM) phenomenon [15] where the natural variation in a sampled population can, by chance, look like a real change but subsequent samples are back closer to the mean.

#### 3.1.3. Weight and Feeding Behaviour

There were no obvious physiological or behavioural changes observed in the crayfish during Exp. 1. All crayfish recovered quickly after the injection and bleeding procedures. Except for one crayfish, they survived the injection. On weighing the crayfish before and after 15 days of administering the injections, no significant change was observed (*p* > 0.05). The weights of the crayfish were not significantly different between the control and experimental group before injections (*p* = 0.614). Furthermore, there was no significant difference in weights before and after the injections for both groups (*p* > 0.05).

### 3.2. Experiment 2

#### 3.2.1. *Chequa iflavirus*

RT-qPCR revealed that the crayfish had cleared *Chequa iflavirus* within 24 hpi (Figure 3). The haemolymph collected at 48 hpi confirmed the absence of *Chequa iflavirus* in the crayfish. The husbandry control (HC) crayfish were also uninfected by the *Chequa iflavirus* based on haemolymph sampled 24 hpi and 48 hpi, suggesting the lateral transfer of *Chequa iflavirus* via water did not occur over this short time. The melt analysis showed that the host crayfish amplicon had an average melting temperature of 85.77 ± 0.31 °C while the *Chequa iflavirus* standards of concentration 10^7^ had an average melting temperature of 80.05 ± 0.1 °C (Figure 3b). The temperature threshold was set to 76.7 °C, and the amplification efficiency for the assay was 0.89. No mortalities were observed during this experiment, and all crayfish recovered from the stressful injections and bleeding. There was no obvious change in their feeding behaviour, and all crayfish looked healthy upon visual inspection.

#### 3.2.2. Athtab Bunyavirus

Athtab bunyavirus copies did not differ significantly over four bleedings in any treatment (*p* = 0.5351). In the viral inoculum group, one crayfish (I4) had a noticeable decrease in copy numbers for the 48 hpi bleed when compared to other bleeds. For all other samples in both groups, the viral load remained between 10^7^ and 10^8^ copies/µL (Figure 4a,b). Furthermore, for all the samples, Athtab bunyavirus copies/µL were apparently lower in the last bleed (after 14 d) when compared to the first bleed (after 24 h), but it was not statistically significant (Figure 4a,b).

### 3.3. Histopathology

Histology of the negative control and viral injection crayfish revealed that the gross muscle structure of all the crayfish was normal, unperturbed by haemolytic infiltrations or hyperchromatosis. Furthermore, no crayfish had prominent infections by bacteria or other secondary pathogens, apart from Athtab bunyavirus. This was supported by the fact that there was an absence of granulomas and, therefore, bacteria around the haemolymph vessels in the hepatopancreatic tissue (Figure 5a), c.f., Hayakijkosol and Owens [16]. All crayfish except the ones injected with virus-containing haemolymph showed convoluted hepatopancreatic lumens with high columnar tissue, indicating they were assimilating sufficient nutrients. Only in crayfish injected with the virally infected haemolymph were haemocytic infiltration (Figure 5b), melanisation and necrosis evident in the primary filament of the gills (Figure 5b). Some samples also had melanisation in the cuticle of the secondary gill filaments. Karyopyknosis was observed in some cells within the secondary filament, and the formation of “signet ring cells” due to the peripheral migration of chromatin was consistent with transferred infections of putative gill parvovirus [17] (Figure 5c), clearly demonstrating the infectious nature of this “signet ring” phenomenon.

## 4. Discussion

The importance of achieving a disease-free or disease-resistant stock of farmed crustaceans has implications for their sustenance on the farm as well as on their marketability. The rapid clearing (<24 h) of *Chequa iflavirus* by redclaw crayfish seen in the present study is reminiscent of the clearing of Bohle *Ranavirus* by the same species reported by Field [18]. The evidence of a group of redclaw crayfish that resist infections by a virus associated with large-scale mortalities gives aquaculturists a chance to utilise them to generate higher and healthier yields in the future.

As most pathogens have carbohydrate-based surfaces composed of liposaccharides, 1,3-glucan or peptidoglycans, crustaceans have had to create a separate protein-based immune system to deal with viruses [19]. It is hypothesised that the quick response generated by the crayfish (<14 days Exp. 1 and <24 h in Exp. 2) was due to the interfering RNA pathway (iRNA) and the downstream activation of phagocytosis and as seen in this study, melanisation linked to pattern recognition by possibly *Cq*DSCAM. The iRNA mechanism has been proven to reduce viral titre in crustaceans [20] and reduce mortalities in crayfish infected with *Macrobrachium rosenbergii* nodavirus [16] and shrimp infected with WSSV [21]. As the *Chequa iflavirus*-free crayfish used in this study were derived from a population infected with the virus, the hypothesis of heritable anti-viral immunity could explain why these crayfish could eliminate the virus without significant mortalities [17].

The role of *Cq*DSCAM could be pivotal in the fast clearance and prolonged immunity against *Chequa iflavirus*. This protein can be soluble as an opsonin or membrane-bound on immune cells in arthropods, both forms triggering phagocytosis of the pathogen [22,23,24,25]. Studies have found that *Cq*DSCAM provides prolonged protection against WSSV after initial infection, and *Cq*DSCAM expression increases and lasts longer after a second infection with the virus [26,27]. In the present study, clearance of *Chequa iflavirus* within 24 h in Exp. 2 could be an effect of the first exposure to the virus in Exp. 1. The phagocytosis and melanisation, seen herein only in the virus-injected crayfish, could have been accelerated by pathogen-specific binding of *Cq*DSCAM, which would have allowed it to avoid iflaviral infections even when the crayfish were housed for a long duration with infected crayfish. Indeed, the transcriptome of crayfish from the donor population and from the population from which the resistant crayfish came produced over 60 variants of *Cq*DSCAM, demonstrating that the system is active in these crayfish.

A load of Athtab bunyavirus did not change significantly and did not cause any disease throughout the study, even after additional bunyavirus was injected; in addition to the fact that all the crayfish used in the present study were already infected by Athtab bunyavirus but with no signs of disease, this can be explained by the viral accommodation theory [28,29]. The theory suggests that crustaceans such as redclaw crayfish can actively “accommodate” viruses without severe disease or mortality via viral-triggered apoptosis. The viruses persistently infect the host and act as a “memory” for immune responses against future infections by the same virus and to keep the viral load in check [28]. The theory also suggests that when a superinfection occurs, the “resident” virus can somehow provide protection against a second, unrelated virus and thus lower overall mortality and disease caused by the second virus [29], perhaps by the anti-viral system being up-regulated. As in this case, the experimental crayfish had Athtab bunyavirus, which persistently infected redclaw crayfish, and thus, by the theory of viral accommodation, it could have had a role to play in the clearance of *Chequa iflavirus*. However, inferences should be made with caution as this study acknowledges the small sample size of crayfish.

No observable changes in feeding behaviour, weight or physiology of the crayfish were observed. The tank scavengers and housing-control crayfish remained uninfected by *Chequa iflavirus*. Although one mortality was observed in the viral inoculum group, crayfish were found to be free of infections by *Chequa iflavirus* and did not have significantly different Athtab bunyavirus titre when compared to other crayfish. This mortality could be the result of uncontrolled apoptosis of infected cells. Apoptosis is a host defence strategy to limit viral replication, and to overcome this defensive system, viruses have developed numerous mechanisms, including, but not limited to, altering the biochemistry of the cells they infect and inhibiting enzymes involved in apoptosis [19,29,30,31]. However, in case of an environmental perturbation, as experienced by the donor crayfish, the crayfish are unable to contain the viral infections, and their immune system fails to control the apoptotic mechanism, leading to mortality [28,31].

Overall, the study provides evidence for redclaw crayfish being resistant to infections by *Chequa iflavirus*, but future studies would need to incorporate crayfish previously unexposed to *Chequa iflavirus* to truly understand the pathogenicity of the virus. Unfortunately, to date, no virus-free population of redclaw crayfish has been found [12]. In addition, after the current *Chequa iflavirus*-free population held at JCU grows to an adequate population size, a future study can be designed with a larger sample size. This would allow for more robust inferences to be made from the experiments and will increase statistical power considerably. As suggested by this study, the immune responses demonstrated by the crayfish are key to understanding the mechanism of viral clearance. However, to implicitly state which immune response had a bigger role to play would require gene regulation studies or bioassays, including the use of gene probes. Future studies can be designed to discern the role of iRNA or *Cq*DSCAM and can target the anti-apoptotic mechanism controlled by the viruses.

## Figures and Tables

**Figure 1 microorganisms-11-00578-f001:**
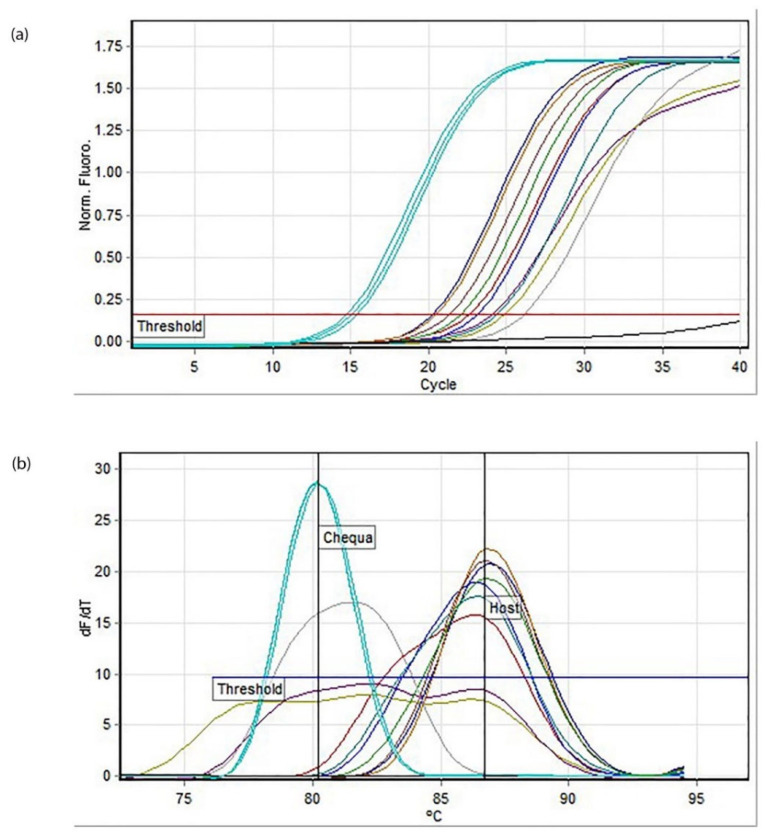
RT-qPCR assay for *Chequa iflavirus* in all crayfish samples post-injections in Experiment 1: (**a**) amplification plot, (**b**) melt analysis (Mean 86.7, SD 0.25). Blue represents *Chequa iflavirus* standards (10^7^), and other colours represent the crayfish samples.

**Figure 2 microorganisms-11-00578-f002:**
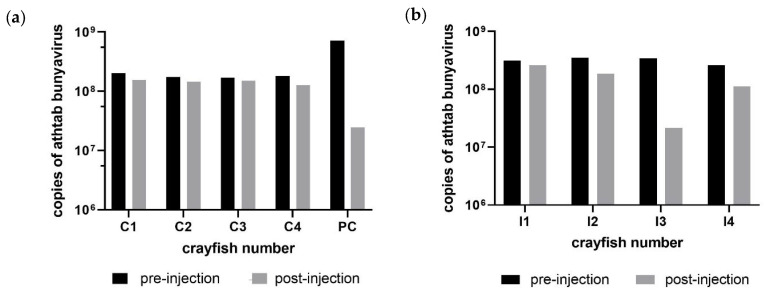
(**a**) Difference in copy number of Athtab bunyavirus before and after 14 days of PBS injections in the control and in-contact, procedural control (PC) group for Experiment 1. (**b**) Difference in copy number of Athtab bunyavirus before and after 14 days in the virus injection group for Experiment 1.

**Figure 3 microorganisms-11-00578-f003:**
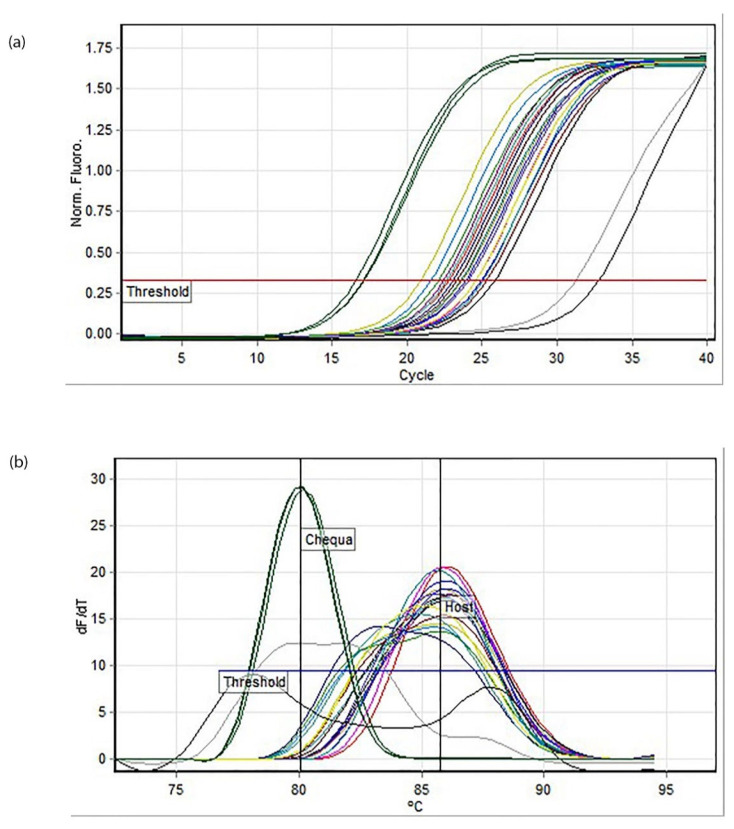
RT-qPCR assay for *Chequa iflavirus* in all crayfish samples post-injections in Experiment 2: (**a**) amplification plot, (**b**) melt analysis (Mean 85.77, SD 0.31). Green represents *Chequa iflavirus* standards (10^7^), and other colours represent the crayfish samples.

**Figure 4 microorganisms-11-00578-f004:**
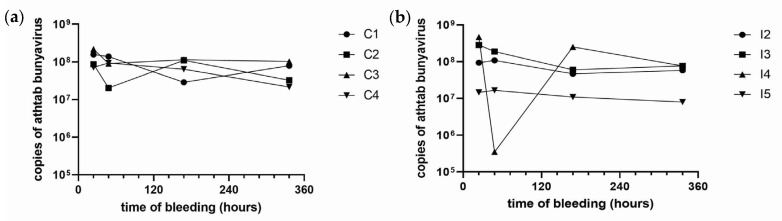
(**a**): Difference in copy number of Athtab bunyavirus from 4 bleedings in the control group: Bleed 1, 24 h: Bleed 2, 48 h: Bleed 3, 7 d: Bleed 4, 14 d. (**b**): Difference in copy number of Athtab bunyavirus from 4 bleedings in the viral injection group: Bleed 1, 24 hpi: Bleed 2, 48 hpi: Bleed 3, 7 dpi: Bleed 4, 14 dpi.

**Figure 5 microorganisms-11-00578-f005:**
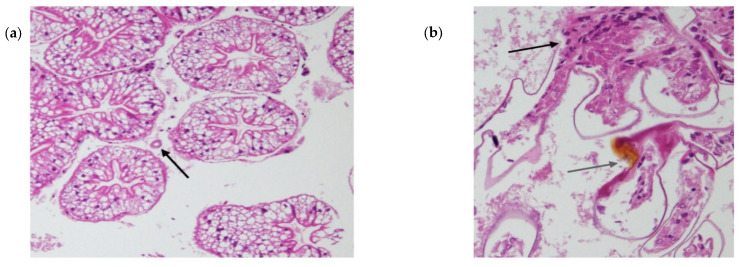
Histology of crayfish tissues. (**a**) granuloma-free haemolymph vessel in the hepatopancreatic tissue of infected crayfish. (**b**) black arrow: aggregation of nuclei (inflammation and tissue repair); grey arrow: necrosis and melanisation in infected crayfish. (**c**) signet ring cells in the secondary gill filament of infected crayfish. Note the large area of melanisation at the right of the photograph.

**Table 1 microorganisms-11-00578-t001:** Primers used to detect *Chequa iflavirus* (Ch) and Athtab bunyavirus (Bu) using RT-qPCR.

	Primer Name	Sequences (5′–3′)	References
1.	Ch-864F	CTCCTTCTGGGTGCGCTTTA	[10]
2.	Ch-976R	ATACTCTGGCGCATGCTCTC	[10]
3.	Bu-2889F2	GATCCGGCAGAATACGAGGG	[11]
4.	Bu-3095R2	ACAACTGTCTGGCTACTGGC	[11]

**Table 2 microorganisms-11-00578-t002:** Design for Experiment 1.

Tank	Treatment (Injection)	Crayfish Number
Control group (C)	PBS	C1, C2, C3, C4
Experimental group	Viral extract	I1 I2, I3, I4
	PBS (Procedural control)	PC
	No injection (Husbandry control)	HC

**Table 3 microorganisms-11-00578-t003:** Design for Experiment 2.

Tank	Treatment (Injection)	Crayfish Number
Control group (C)	PBS	C1, C2, C3, C4
Experimental group	Haemolymph with viruses	I2, I3, I4, I5 (I5 was HC in Exp. 1)
	No injection (Husbandry control)	HC (Previously PC in Exp. 1)

## Data Availability

Data can be requested from the corresponding author.

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
