# Peer review of "Cherax quadricarinatus Resistant to Chequa iflavirus: A Pilot Study"

_microorganisms, 2023, doi:10.3390/microorganisms11030578_

Round 1

Reviewer 1 Report

The main critique of the paper is the mention of DSCAM immunity without experimental evidence. Are any molecular markers or microsatellites to be used to characterize this resistant group?

References on molecular markers:

MAX-AGUILAR, Adriana et al. Genetic diversity of divergent red claw crayfish Cherax quadricarinatus (Von Martens, 1868) populations evaluated to initiate a breeding program in Mexico. Lat. Am. J. Aquat. Res. [online]. 2021, vol.49, n.2   http://dx.doi.org/10.3856/vol49-issue2-fulltext-2630.

Population genetic studies on the Australian freshwater crayfish, Cherax destructor (Crustacea: Parastacidae) using allozyme and RAPD markers. Thuy T. T. NguyenChristopher P. Burridge and Christopher M. Austin. Aquat. Living Resour., 18 1 (2005) 55-64 DOI: https://doi.org/10.1051/alr:2005005

Please include in supp. materials the gel mentioned in line 230.

Also, it would be possible to determine the reason for the death of crayfish subject I1 mentioned in lines 226-228.

Please review other works on the role of DSCAM in the immune response (lines 327-335), like the paper of Ng et al (2919) and this reference:

A Comprehensive Review on Crustaceans’ Immune System With a Focus on Freshwater Crayfish in Relation to Crayfish Plague Disease. Bouallegui, Y. 2021. Frontiers in Immunology 12,667787

Please add doi web links to all references.

Author Response

The main critique of the paper is the mention of DSCAM immunity without experimental evidence.

The abstracts hypothesises both iRNA and DSCAM may be involved.  This references the discussion where the literature regarding these mechanisms in crustaceans as well as viral accommodation theory could relate to our observations. While we suggest these mechanisms could be responsible, we also acknowledge that much more work would need to be done to confirm or reject these hypotheses

Are any molecular markers or microsatellites to be used to characterize this resistant group
References on molecular markers:

MAX-AGUILAR, Adriana et al. Genetic diversity of divergent red claw crayfish Cherax quadricarinatus (Von Martens, 1868) populations evaluated to initiate a breeding program in Mexico. Lat. Am. J. Aquat. Res. [online]. 2021, vol.49, n.2   http://dx.doi.org/10.3856/vol49-issue2-fulltext-2630.

Population genetic studies on the Australian freshwater crayfish, Cherax destructor (Crustacea: Parastacidae) using allozyme and RAPD markers. Thuy T. T. Nguyen, Christopher P. Burridge and Christopher M. Austin. Aquat. Living Resour., 18 1 (2005) 55-64 DOI:
https://doi.org/10.1051/alr:2005005

We do plan to do genetic characterisation prior to outbreeding experiments and thank the reviewer for references provided, these will be utilised as part of our planning

Please include in supp. materials the gel mentioned in line 230.

Unfortunately the original digital file has been corrupted and the print out is not at journal quality resolution.  This gel is supplementary to the original RTqPCR results which are included and present the same data.

Also, it would be possible to determine the reason for the death of crayfish subject I1 mentioned in lines 226-228.

It is not possible to determine this more thoroughly than we have already – see line 371-374 in the discussion.  We have excluded an increase in viral load and speculate this due to uncontrolled apoptosis.

Please review other works on the role of DSCAM in the immune response (lines 327-335), like the paper of Ng et al (2919) and this reference:

A Comprehensive Review on Crustaceans’ Immune System With a Focus on Freshwater Crayfish in Relation to Crayfish Plague Disease. Bouallegui, Y. 2021. Frontiers in Immunology 12,667787

Bouallegui has now been included ( line 344 - basis of DSCAM list of references), Ng 2019 has been used in line 346

Please add doi web links to all references.

Done where these exist

Reviewer 2 Report

Manuscript on Nambiar et al refers on proposing a new solution on resistance on viral infections on aquaculture production on the redclaw crayfish (Cherax quadricarinatus). Manuscript on well-written and on presented lacks some additional clarifications. First on routes on exposure and transmission on the two viral particles? Is water or temperature a vehicle on transmission on aquaculture production? How does Cherax quadricarinatus gets infected? On via humans or water? Another remark refers on why were the two virus studied together? I mean on most frequent or on previously considered together? There should be a description on other studies on these viruses on the Introduction section. Regarding RT-PCR on viral counts represent active viral particles?

Finally, on experiments the cause attributed on lower mortality on acquired immune resistance. On so can this alter on water parameters on aquaculture production and on how impactful on resistance on next generations on Cherax quadricarinatus?  

Author Response

Manuscript on Nambiar et al refers on proposing a new solution on resistance on viral infections on aquaculture production on the redclaw crayfish (Cherax quadricarinatus). Manuscript on well-written and on presented lacks some additional clarifications. First on routes on exposure and transmission on the two viral particles? Is water or temperature a vehicle on transmission on aquaculture production?  How does Cherax quadricarinatus gets infected? On via humans or water?

The paper references the limited knowledge regarding route of exposure – both these viruses can be transferred in ovo (line 58 Jaroenram et al 2021).  As all populations tested have Athtab, it is not possible to test for other routes of infection.  This experiment was to test via injection (and by co-location in the same water) whether chequa could be transferred prior to using other methods.  The effect of temperature stress has not yet been studied with this virus.  There are no examples of any zoonotic virus between humans and crayfish, however it is certainly possible (generally) for diseases to spread between crayfish via humans moving crayfish from an infected group to a non-infected group.

Another remark refers on why were the two virus studied together? I mean on most frequent or on previously considered together? There should be a description on other studies on these viruses on the Introduction section. Regarding RT-PCR on viral counts represent active viral particles?

The two viruses are studied together because all available crayfish are endemically infected with Athtab bunyavirus (donor and chequa free animals), so monitoring of this has to be done – this is noted in  lines 56-58.   The introduction does mention the previous research on these viruses (isolation, in ovo, transmission).  Viral counts are expected to represent active particles as the virus has an RNA genome, as this is less stable than DNA. Experiment 2 was designed to eliminate the possibility that the extraction process inactivated the particles by directly injecting fresh haemolymph from infected animals in population 1

Finally, on experiments the cause attributed on lower mortality on acquired immune resistance. On so can this alter on water parameters on aquaculture production and on how impactful on resistance on next generations on Cherax quadricarinatus? 

The crayfish tested are the offspring of the original chequa free animals, indicating that the resistance is multigenerational (see line 65-66).  Cross breeding with infected animals is planned but not yet done. 

Reviewer 3 Report

The manuscript microorganisms-2207697 describes the resistance to Chequa iflavirus of Cherax quadricarinatus. The resistance is highlighted by means of experimental tests in the microcosm.

The paper is certainly interesting, but it is not always easy to use. In fact, both the materials and methods and the results are not well articulated and sufficiently described; the number of subjects used in the two trials is not explicit, even if it can be inferred from the following text. However, the number used is in my opinion insufficient to be able to make the assertions that the authors make in the manuscript.

Furthermore, the bibliography must be completely reviewed, in part by inserting it as the journal requires and in part by checking that all the citations made in the text correspond to those in the references. For example, the citations to lines 208-209 and 216 are missing from the references; on the other hand, the citations on lines 408-409, 426-427, 435-437, 448-450, 453-455, 456-457, 460-462, 463, 475-476 and 477-478 are missing from the text. Indeed, it is necessary to check the entire bibliography formatting it as required (ascending numbering in the text and insertion in numerical order in the references).

For these reasons, in my opinion this paper cannot be accepted in its present form.

Author Response

The manuscript microorganisms-2207697 describes the resistance to Chequa iflavirus of Cherax quadricarinatus. The resistance is highlighted by means of experimental tests in the microcosm.

The paper is certainly interesting, but it is not always easy to use. In fact, both the materials and methods and the results are not well articulated and sufficiently described; the number of subjects used in the two trials is not explicit, even if it can be inferred from the following text. However, the number used is in my opinion insufficient to be able to make the assertions that the authors make in the manuscript.

The methods section has been significantly rewritten to identify the number and disposition of crayfish used in the research.  Specifically, section 2.1 (crayfish ) has been rewritten to focus on the number and sources of crayfish while allocation of crayfish is described in experimental design (2.6.2 and 2.6.3).  The separate sections of the methodology reference each other where this aids in clarity

The limited number of animals has been acknowledged, as has the fact that this limited number requires any conclusion or inference to be considered cautiously.  This has been done by having “pilot study” in the title, explaining the reason why only a small number of animals was available (line 65-66) and qualifying the rigor of the conclusions given the small size and requirement for a more thorough investigation once the population expands.(line 384-388)

Furthermore, the bibliography must be completely reviewed, in part by inserting it as the journal requires and in part by checking that all the citations made in the text correspond to those in the references. For example, the citations to lines 208-209 and 216 are missing from the references; on the other hand, the citations on lines 408-409, 426-427, 435-437, 448-450, 453-455, 456-457, 460-462, 463, 475-476 and 477-478 are missing from the text. Indeed, it is necessary to check the entire bibliography formatting it as required (ascending numbering in the text and insertion in numerical order in the references).

For these reasons, in my opinion this paper cannot be accepted in its present form.

The reference section has been thoroughly reviewed and all references in the paper matched to those in the reference list.  The referencing has been changed to use ascending numbering in text and numerical format in the reference list

Round 2

Reviewer 2 Report

Authors on adressed comments and now manuscript warrants publication on Microorganisms.

Author Response

Thankyou for your time with this manuscript

Reviewer 3 Report

The manuscript microorganisms-2207697 has been revised as requested, certainly obtaining a better use of the text which is always very difficult to understand.

The experimental part was clarified by the authors with an easier use. The citations have been inserted correctly; there is still an oversight in lines 101 and 107 where the dates should be deleted (2017b).

For these reasons and for the interest of the topic, the paper in its current form can be published after minor revision.

Author Response

Thank you for your comment.  The 2017b dates have been removed.